# Artificial Intelligence for Electrochemical Prediction and Optimization of Direct Carbon Fuel Cells Fueled with Biochar

Adam Cherni [1] and Kamel Halouani [2,3,*]

1 Tunisia Polytechnic School (EPT), University of Carthage, V8HP+VXW, Rue El Khawarizmi, Site Archéologique de Carthage, P.O. Box 743, La Marsa 2078, Tunisia; adam.cherni@ept.ucar.tn

2 Laboratory of Systems Integration and Emerging Energies (LR21ES14), National Engineering School of Sfax (ENIS), University of Sfax, IPEIS, Road Menzel Chaker km 0.5, P.O. Box 1172, Sfax 3018, Tunisia

3 Smart Bio-Energy Systems, Digital Research Center of Sfax, Technopole of Sfax, Sakiet Ezzit, P.O. Box 275, Sfax 3021, Tunisia

* Correspondence: kamel.halouani@ipeis.usf.tn; Tel.: +216-98-954-415

**Abstract:** At present, direct carbon fuel cells constitute an emerging energy technology that electrochemically converts solid carbon to electricity with high efficiency. The recent trend of DCFCs fueled with biochar from biomass carbonization as green fuel has reinforced the environmental benefits of DCFCs as a clean and sustainable technology. However, there remain new challenges related to some complex unknown kinetic parameters, $X = (\alpha_a, \alpha_c, \sigma_g, i_{0,a}, i_{0,c}, i_{l_{O_2}}, i_{l_{CO_2,c}}, i_{l_{CO_2,a}}, i_{l_{CO}})$, of the electrochemical conversion of biochar in DCFCs and there is a need for intelligent techniques for prediction and optimization, refering to the available experimental data. The differential evolution (DE) algorithm, which ranked as one of the top performers in optimization competitions with competitive accuracy and convergence speed, was used here for providing the optimized values of these parameters by minimizing the root mean squared errors (RMSE). The proposed technique was then applied to DCFCs fueled by activated pure carbon (APC) using $CO_2$ and $CO/CO_2$ electrochemical models with RMSE around $10^{-2}$ and $10^{-3}$, respectively. Then, the $CO/CO_2$ model was applied to a DCFC fueled with almond shell biochar (ASB), which displayed a slight increase in RMSE (of the order of $10^{-2}$) due to the complex porous structure of ASB and the content of additional chemical elements that affect the electrochemistry of the DCFC and are not considered in the model.

**Keywords:** direct carbon fuel cell (DCFC); biochar; electrochemical parameters; artificial intelligence; differential evolution (DE) algorithm; prediction; optimization; experimental validation

## 1. Introduction

Across the world, extensive research and substantial investments in renewable energies have been undertaken aiming to secure sustainable sources of energy, effectively tackle the pressing issues of global warming and uphold a high-quality environment for future generations [1]. In this context, biomass stands out as the most prevalent and abundant renewable energy source. Unlike fossil fuels, biomass does not contribute to the greenhouse effect due to its reliance on the natural carbon cycle. This makes it a promising alternative to mitigate climate change impacts. Thanks to its carbon-rich nature [2], biomass and biochar produced from biomass carbonization have been extensively researched for their potential use in direct carbon fuel cells (DCFCs) to produce sustainable electricity [3–6]. Indeed, these cells directly convert the chemical energy of carbonous fuel into green electricity with high efficiency [7–9]. DCFCs can be categorized on the basis of their electrolyte type, such as molten hydroxide, molten carbonate and oxygen/carbon dioxide ions conducting ceramic. Additionally, various sub-categories of DCFCs exist, offering versatility through different anode materials, anode chamber designs and fuel delivery methods [7]. The implementation of DCFCs provides several advantages over conventional fuel cells and power generation technologies. These benefits include high net efficiency

approaching 70%, the potential for almost sequestration-ready $CO_2$ emissions without significant additional costs or energy losses for capture and the utilization of low-cost fuel sources [9]. Elleuch et al. [10] conducted comprehensive research on DCFCs focused on $CO_2$ and $CO/CO_2$ production. These authors successfully developed and validated an analytical model for $CO_2$-producing fuel cells at 923 K. Additionally, they explored $CO/CO_2$ DCFC behavior characteristics under different operating temperatures (923 K and 1023 K). The investigation also encompassed the evaluation of the $CO_2/O_2$ ratio and cathode pressure's influence on the system's performance. Their findings indicated that higher temperatures contributed to enhancing the system performance. Furthermore, in separate studies, Elleuch et al. [5,6] explored the use of biochar as fuel for DCFCs across various temperature settings. They particularly examined the production of CO and $CO_2$ from almond shell biochar at the anode side and evaluated the cell's power and voltage performance under different operating conditions, such as temperature variations and the utilization of activated biochar. The available experimental data showed good DCFC performance with ASB as fuel but it is difficult to analyze DCFCs' real-time behavior due to their complex and hybrid material composition, non-linear features, substantial hysteresis behavior and insufficient datasets, making it hard to optimize and control its several unknown electrochemical parameters, particularly when fueled with ASB.

Indeed, there are several gaps in the knowledge regarding electrochemical conversion mechanisms of biochar and the transfer phenomena in the complex microstructure of the different DCFC zones, knowing that realistic implementation remains a challenge.

This challenge can be overcome by using AI algorithms for the resolution of such multivariable complex nonlinear problems, allowing for the prediction and optimization of these key operating parameters of DCFCs using the available experimental data [5,6,10].

Indeed, nowadays, AI algorithms are increasingly used in fuel cell applications as predictive models, as they are reliable and efficient tools for the optimization of fuel cell design and operating parameters. Artificial neural networks (ANN), support vector machines (SVM) and random forests (RF) have been elaborated on and applied in fuel cells performance prediction and optimization. The review of Su et al. [11] showed that machine learning (ML) algorithms can be successfully used to predict the performance, fault diagnosis and service life of fuel cells. ML can also be used with high accuracy in solving nonlinear problems. They combined optimization algorithms with ML models and carried out the optimal design and operating conditions of proton exchange membranes (PEMFCs) and solid oxide (SOFC) fuel cells to achieve multi-optimization objectives with good accuracy.

Abdollahfard et al. [12] used microbial fuel cell (MFC) datasets to make models using three key parameters (DS, Pt and Aeration) as inputs and power density and/or chemical oxygen demand (COD) removal as outputs. Random forest regression (RG) and gradient boost regression tree (GBRT) algorithms were used to build the MFC machine learning model for the prediction of power density and COD removal. They found the optimal input parameters that maximize power density or COD removal with high accuracy through models using particle swarm optimization.

Kishore et al. [13] have shown great promise for AI algorithms in providing accurate diagnoses of fuel cells through rapid data collection. These authors focused on the following common software models: random forest (RF), genetic algorithm (GA), artificial neural network (ANN), particle swarm optimization (PSO), extreme learning machine (ELM) and support vector machine (SVM), in order to conduct a proper and dynamic analysis of fuel cells. According to these authors, these methods are not only popular and useful tools for simulating the nature of fuel cells system, but they are also suitable for optimizing the operational parameters needed for an ideal fuel cell device.

The recent literature [14–20] has shown that using metaheuristic-based techniques can be so useful for solving such complex nonlinear problems. Indeed, metaheuristics are computational intelligence paradigms used especially for the sophisticated solving of optimization problems. They are powerful problem-solving approaches that offer effective

and flexible solutions to complex optimization problems. Metaheuristics are unlike exact algorithms that struggle with large-scale, real-world challenges; metaheuristics excel in finding near-optimal solutions within reasonable time frames. Their strength lies in their ability to explore vast solution spaces and escape local optima, making them highly adaptable and applicable to a wide range of problems. With minimal problem-specific knowledge required, metaheuristics are easy to implement and can be tailored to various domains. Different metaheuristics were used in the field of renewable energy such as PSO [14], artificial bee colony (ABC) [15], whale optimization algorithms (WOA) [16], genetic algorithm (GA) [17] and ant colony optimization (ACO) [18]. Azar et al. [19] used the Battle Royale Optimization Algorithm (DBRA) to effectively identify the undetermined parameters in solid oxide fuel cell models. Additionally, they validated the model's accuracy by comparing it with experimental data for voltage and power under various pressure conditions. Mahdinia et al. [20] applied the CCAB algorithm to optimize the proposed parameters while investigating the system's total cost as a function of temperature, pressure, current density and efficiency. By utilizing this advanced algorithm, the study effectively harnessed the interplay between these crucial variables, offering valuable insights into enhancing the overall system's performance and minimizing costs.

Thanks to its robustness, the DE algorithm demonstrated remarkable resilience in tackling various problem types, which can be multimodal, noisy, or encompassing a multitude of dimensions [21,22]. This adaptability is vital when navigating the intricate behavior of polarization losses estimation in DCFCs. Furthermore, DE's simplicity is a notable asset, characterized by its minimal control parameters, primarily the population size, mutation and crossover scaling factors. This streamlined setup eases the algorithm selection process and reduces the need for extensive parameter fine-tuning, thus streamlining the parameter estimation process [23–25]. In cases where objective functions are noisy or subject to uncertainty, such as those influenced by experimental errors, DE's robustness proves indispensable by aiding in the identification of optimal solutions amidst the noise. Additionally, DE is particularly efficient in the optimization of high-dimensional spaces, making it suitable for tasks involving a multitude of variables like the estimation of polarizations in DCFCs within a large-dimensional parameter space, as is the case in our study [26,27].

In this work, we propose the use of Differential Evolution (DE) algorithm for the prediction and optimization of the following kinetic electrochemical key parameters of DCFCs: the anodic and cathodic charge transfer coefficients ($\alpha_a$, $\alpha_c$), the global DCFC conductivity $\sigma_g$; the anodic and cathodic exchange current densities ($i_{0,a}$, $i_{0,c}$); and the limit current densities of gas spices through the electrodes ($i_{l_{O_2}}$, $i_{l_{CO_2,c}}$, $i_{l_{CO_2,a}}$, $i_{l_{CO}}$), with reference to APC experimental and analytical available data for both the $CO_2$ and $CO/CO_2$ models. Subsequently, we apply the proposed model to analyze the electrochemical behavior of DCFCs fueled with ASB, ensuring a comprehensive evaluation of its I-V and I-P characteristics and performance.

## 2. Materials and Methods

### 2.1. DCFC Fueled by Activated Pure Carbon (APC) and Almond Shell Biochar (ASB)

As described in Figure 1, the DCFC is based on a ceria-carbonate composite electrolyte (2:1 mol ratio $Li_2CO_3/Na_2CO_3$ eutectic mixture). The anode consists of a mixture of APC/Carbonate or ASB/carbonate in a mass ratio of 1:9. The composite cathode consists of 30/70 wt% composite electrolyte and Lithiated NiO ($LiNiO_2$) powders, respectively [5].

Despite their different compositions and electrochemical behaviors, APC and ASB have porous structures with relatively high specific area and share adsorption capabilities, particularly when activated. Indeed, APC and ASB differ in composition and source. While APC consists solely of carbon atoms, and can be naturally occurring or synthetically produced, ASB is produced from the carbonization of almond shell (a lignocellulosic agricultural by-product) and composed of 71.8% C, 23.8% O, 3.9% H, 0.45% N and 0.04% S [6]. As carbon and oxygen together constitute 95.6% of the total mass of the biochar, our model will consider only these two elements. Previous results [6] have shown that the perfor-

mance of the electrochemical conversion of the ASB in DCFCs was highly dependent on its chemical composition, surface area, mineral matter content and its oxygen-functional groups at the surface, on the edges and within the graphitic structure of the carbon crystal of the ASB, where the degree of coverage of reactive sites is higher. Due to their complex material composition, DCFCs exhibit significant nonlinear electrochemical behavior, especially when fueled with ASB, making the optimization and control of their power generation very difficult.

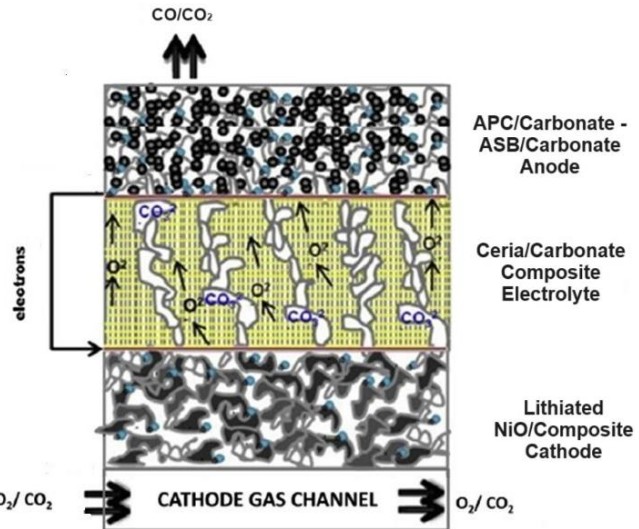

**Figure 1.** Schematic principle of DCFC fueled with APC/ASB.

### 2.2. $CO_2$ and $CO/CO_2$ DCFC Models

$CO_2$ and $CO/CO_2$ producing DCFCs with molten carbonate electrolytes are advanced fuel cell technologies operating at high temperatures. They efficiently convert solid carbon-based fuels into electricity through oxidation at the anode, resulting in $CO_2$ production. Additionally, some designs can electrochemically reduce $CO_2$ at the anode, generating CO [10,28–33]. In particular, when ASB is used as fuel in DCFCs, CO is the dominant gas at the anode [5].

The $CO_2$ model can be described using the following electrochemical electrode reactions:

$$\text{Anode reaction}: \ C + 2CO_3^{2-} \rightarrow 3CO_2 + 4e^- \tag{1}$$

$$\text{Cathode reaction}: \ O_2 + 2CO_2 + 4e^- \rightarrow 2CO_3^{2-} \tag{2}$$

$$\text{Overall reaction}: \ C + O_2 \rightarrow CO_2 \tag{3}$$

The $CO/CO_2$ model can be described using the following electrochemical electrode reactions:

$$\text{Anode reaction}: \ C + CO_3^{2-} \rightarrow CO + CO_2 + 2e^- \tag{4}$$

$$\text{Cathode reaction}: \ \frac{1}{2}O_2 + CO_2 + 2e^- \rightarrow CO_3^{2-} \tag{5}$$

$$\text{Overall reaction}: \ C + \frac{1}{2}O_2 \rightarrow CO \tag{6}$$

The Nernst equation determines the ideal potential of a $CO_2$ and a $CO/CO_2$ producing DCFC and depends on the partial pressure of the present gaseous species. For the $CO_2$ model, it is expressed as follows [28,34,35]:

$$E_T = E_T^0 + \frac{RT}{4F} \times \ln\left(\frac{P_{O_2}P_{CO_2,c}^2}{P_{CO_2,a}^3}\right) \tag{7}$$

For the $CO/CO_2$ model, it is expressed as follows [30,36–38]:

$$E_T = E_T^0 \frac{RT}{n_e F} \left( n_c \ln\left(\frac{P_c}{P_{anode}}\right) + n_{O_2}\ln\left(\frac{P_{O_2}}{P_{cathode}}\right) + n_{CO_2,c}\ln\left(\frac{P_{CO_2,c}}{P_{cathode}}\right) + n_c\ln\left(\frac{P_{CO_2,a}}{P_{anode}}\right) - n_{CO_2,c}\ln\left(\frac{P_{CO_2,a}}{P_{anode}}\right) \right)$$
$$-2n_{O_2}\ln\left(\frac{P_{CO_2,a}}{P_{anode}}\right) - 2n_c\ln\left(\frac{P_{CO}}{P_{anode}}\right) + 2n_{O_2}\ln\left(\frac{P_{CO}}{P_{anode}}\right))$$

(8)

At low current densities in a $CO_2$ producing DCFC, activation polarization plays a significant role. This polarization arises from the electrochemical reaction at the carbon–molten carbonate electrolyte interfaces, where the reactants need to overcome an energy barrier known as the activation energy in order to facilitate electron transfer. The activation polarization is accurately described by the Butler–Volmer equation:

$$i = i_{0,a} \exp\left(\frac{\alpha_a \eta_{act,a} n_e F}{RT}\right) - i_{0,a} \exp\left(\frac{-(1-\alpha_a)\eta_{act,a} n_e F}{RT}\right)$$

(9)

$$i = i_{0,c} \exp\left(\frac{\alpha_c \eta_{act,c} n_e F}{RT}\right) - i_{0,c} \exp\left(\frac{-(1-\alpha_c)\eta_{act,c} n_e F}{RT}\right)$$

(10)

Therefore, the activation losses are given by the following:

$$\text{Anode activation}: \quad \eta_{act,a} = \frac{RT}{\alpha_a n_e F} \ln\left(\frac{i}{i_{0,a}} + 1\right)$$

(11)

$$\text{Cathode activation}: \quad \eta_{act,c} = \frac{RT}{\alpha_c n_e F} \ln\left(\frac{i}{i_{0,c}} + 1\right)$$

(12)

where $\alpha_a$ and $\alpha_c$ are, respectively, the charge transfer coefficients in the anode and cathode serve as dimensionless parameters that elucidate the symmetry of electrochemical reactions related to electron transfer. The determination of these coefficients often involves experimental methods such as electrochemical impedance spectroscopy for the anodic coefficient or cyclic voltammetry and chronoamperometry for the cathodic coefficient. Additionally, fitting experimental data to the Butler–Volmer equation is a common approach to ascertain these crucial parameters, giving arbitrary values between 0 and 1 (depending on the symmetry of the transition state in the electrochemical reactions). In this work, we propose the prediction of the optimal near-real values of these coefficients using new AI tools.

According to Arrhenius law, the exchange current density ($i_{0,a}$) at which the rate of the forward reaction (oxidation) equals the rate of the reverse reaction (reduction) at the anode under equilibrium conditions is as follows:

$$i_{0,a} = K_B \exp\left(\frac{-E_B}{T}\right)$$

(13)

The pre-exponential factor of the backward reaction is $K_B = 5.8 \times 10^9$ A·m$^{-2}$ and the activation energy of the backward reaction is $E_B = 22.175$ K$^{-1}$ [36].

The anodic exchange current density is influenced by several operating factors such as the temperature, the anode material and the specific electrochemical reaction taking place on the anode side.

The cathodic exchange current density ($i_{0,c}$) is the rate at which the backward reaction (reduction) equals the rate of the forward reaction (oxidation) at the cathode side under equilibrium conditions. As far as the anodic exchange current density, the cathodic exchange current density is also affected by the operating factors like the temperature, the cathode material and the specific electrochemical reaction occurring at the cathode. As multi-parameters and nonlinear electrochemical variables, $i_{0,a}$ and $i_{0,c}$ cannot be predicted accurately using classical numerical methods. An advanced identification AI algorithm and modern computational approaches may be used to anticipate the properties of the complex materials as well as the optimization of the DCFC process.

Ohmic polarization emerges as a result of resistance encountered by ions and electrons within the single domain cell, impeding their flow. This polarization follows Ohm's law and is mathematically expressed by the Equation (14) [32–37]:

$$\eta_{ohmic} = R_{ohmic} \times i \tag{14}$$

$$\text{Where: } R_{ohmic} = \frac{\delta_g}{\sigma_g} \tag{15}$$

where $\delta_g = 10^{-3}$ m and $\sigma_g$ (S·m$^{-1}$) are, respectively, the thickness and the global conductivity of the cell.

The global electrical conductivity of a cell ($\sigma_g$) in the context of ohmic polarization refers to the overall ability of the cell to conduct electrical current and ions in electrodes and electrolytes, respectively. It depends on the materials of the electrodes and electrolytes and the operating temperature. Ohmic polarization occurs when the dominant factor limiting the flow of the current in an electrochemical cell is the resistance of the electrolyte.

Concentration polarization in a $CO_2$ producing DCFC can arise from the slow diffusion of gas species (i.e., $CO_2$, $O_2$) from the cathode inlet to the reaction zone, from the motion of un-reacted $O_2$ and $CO_2$ from the cathode and from the diffusion of reactants and products through the electrolyte to and from the electrochemical reaction sites. Although concentration polarization is typically formulated based on thermodynamic principles, Basio et al. [38] demonstrated a more consistent kinetic expression. The concentration polarization in the $CO_2$ producing DCFC is defined using Fick's law as follows:

$$\eta_{conc} = \frac{RT}{\nu_e F} \sum \nu_i \ln\left(\frac{C_i^S}{C_i^B}\right) \tag{16}$$

Therefore, using (1) and (2):

$$\eta_{conc} = \frac{RT}{F} \times \left(\frac{1}{4}\ln\left(1 - \frac{i}{i_{l_{O_2}}}\right) + \frac{1}{2}\ln\left(1 - \frac{i}{i_{l_{CO_2,c}}}\right) - \frac{3}{4}\ln\left(1 - \frac{i}{i_{l_{CO_2,a}}}\right)\right) \tag{17}$$

The $CO/CO_2$ producing DCFC follows the same principle as the $CO_2$ system, with one key difference being that it involves an additional anodic reaction producing a mixture of CO and $CO_2$. Regarding modeling, the $CO/CO_2$ producing DCFC employs the same activation and ohmic losses formulation discussed in the previous section. The only difference is the number of electrons involved, which reduces from four to two due to the concurrent formation of CO and $CO_2$ on the anode side. Hence, the concentration polarization and ideal potential of the $CO/CO_2$ producing DCFC are determined while considering the simultaneous formation of both CO and $CO_2$ on the anode side. The ideal potential of the $CO/CO_2$ producing DCFC is given by Equation (8) and the concentration polarization is given by Equation (18) according to the electrochemical reaction Equations (4) and (5):

$$\eta_{conc} = \frac{RT}{F} \times \left(\frac{1}{4}\ln\left(1 - \frac{i}{i_{l_{O_2}}}\right) + \frac{1}{4}\ln\left(1 - \frac{i}{i_{l_{CO_2,c}}}\right) - \frac{1}{2}\ln\left(1 - \frac{i}{i_{l_{CO_2,a}}}\right) - \frac{1}{2}\ln\left(1 - \frac{i}{i_{l_{CO}}}\right)\right) \tag{18}$$

The limit current densities ($i_{l_{O_2}}$, $i_{l_{CO_2,c}}$, $i_{l_{CO_2,a}}$ and $i_{l,co}$) of the gas spices in the cathode and anode refer to the maximum current densities that can be sustained at an electrode interface when mass transport limitations become the dominant factor. As the current density increases beyond the limit, the concentration of reactants or products near the electrode surface deviates significantly from the bulk concentration, leading to a drop in the current efficiency and other undesirable effects.

Typically, at low current density, the main source of voltage loss in a fuel cell is activation polarization. However, as the current density rises, ohmic polarization takes over as the primary cause of voltage drop. Eventually, at high current densities, concentration

polarization becomes the dominant factor contributing to the voltage loss. It is worth noting that the theoretical output voltage of the fuel cell is as follows:

$$V_{cell} = E_T - \eta_T \tag{19}$$

$$V_{cell} = E_T - \left(\eta_{act,a} + \eta_{act,c} + \eta_{conc,a} + \eta_{conc,c} + \eta_{ohmic}\right) \tag{20}$$

It can be seen that the nonlinear Equation (20) contains several unknown parameters affecting the electrochemical conversion mechanism of the DCFC. Hence, appropriate AI optimization strategies should be used for the effective and accurate prediction of these parameters. Accordingly, the differential evolution (DE) algorithm is proposed here to determine the following seven unknown parameters: the anode charge transfer coefficient ($\alpha_a$), the cathode charge transfer coefficient ($\alpha_c$), the global electrical conductivity of the single-domain DCFC ($\sigma_g$), the cathode exchange current density ($i_{0,c}$) and the limit current densities of oxygen ($i_{l_{O_2}}$), the carbon dioxide on the cathode side ($i_{l_{CO_2,c}}$) and the carbon dioxide on the anode side ($i_{l_{CO_2,a}}$). Taking into account that these seven parameters will considerably affect the DCFC electrochemical behavior, they must thus be accurately estimated to fulfill the actual I–V and I-P characteristics of the DCFC.

### 2.3. Differential Evolution (DE) Algorithm

DE, belonging to the evolutionary algorithm family, generates new solutions by recombining existing ones, making it robust and governed by few algorithm-specific parameters. DE outperforms other optimization methods in tackling challenging problems with nonlinear, multimodal, and non-separable features. It was consistently ranked as one of the top performers in optimization competitions, demonstrating its potential for real-world applications with competitive accuracy and convergence speed. Notably, DE's lower space complexity grants it superior scalability, particularly for handling large-scale and computationally intensive optimization problems. These advantages make DE a favored choice among researchers and practitioners for solving diverse sets of real-world optimization challenges effectively and efficiently [21–27]. Figure 2 illustrates the scheme of DE calculation used in this paper. The DE algorithm consists of four phases which are as follows: initialization; mutation; crossover and evaluation; and population update. In the version used in this paper, we added a boundary refinement phase.

- Population initialization: This is the first process step of DE-generating random individuals within the specified boundaries, which are vectors in a D-dimensional space. Each ith individual solution of DE can be represented as a D-dimensional vector as follows [23]:

$$X_i^t = (X_{i,1}, X_{i,2}, \dots, X_{i,D}) \tag{21}$$

The specified boundaries are given by the following Equations [25]:

$$X_{min} = (X_{min,1}, X_{min,2}, \dots, X_{min,D}) \tag{22}$$

$$X_{max} = (X_{max,1}, X_{max,2}, \dots, X_{max,D}) \tag{23}$$

For each ith DE solution, the jth dimensional component can be initialized by randomly generating as follows [23]:

$$X_{i,j}^{(0)} = X_{min,j} + \text{rand}_{i,j}[0,1](X_{max,j} - X_{min,j}) \tag{24}$$

- Mutation: For each individual in the population, the algorithm selects three distinct individuals (candidates), $X_{i,1}{}^t, X_{i,2}{}^t$ and $X_{i,3}{}^t$, randomly from the population (excluding the current individual) to calculate the mutant vector:

$$X_m^t = X_i^t + SF \times (X_{i,1}^t - X_{i,2}^t) \tag{25}$$

where SF is the scaling factor and $X_i^t \neq X_{i,1}^t \neq X_{i,2}^t \neq X_{i,3}^t$.

- Crossover: A trial vector is created by combining the mutant vector and the current individual. Each element of the trial vector is determined by a crossover operation.

The crossover operation is performed with a probability of a crossover rate [23]:

$$Z_i^t = \begin{cases} X_{m,j}^t & \text{if } rand_{i,j}[0,1] \leq CR \\ \\ X_{i,j}^t & \text{Otherwise} \end{cases} \tag{26}$$

- Boundary refinement: This prevents solutions from going beyond specified limits, avoiding infeasible or unrealistic outcomes:

$$Z_{i,j}^t = \begin{cases} \max\left(Z_{i,j}^t, X_{min,j}\right) \\ \min\left(Z_{i,j}^t, X_{max,j}\right) \end{cases} \tag{27}$$

- Evaluation and population update: The fitness of the trial vector is evaluated using the objective function. If the fitness of the trial vector is better (lower) than the fitness of the current individual, the current individual is replaced by the trial vector [23,27]:

$$X_i^{t+1} = \begin{cases} Z_i^t & \text{if } f(Z_i^t) \leq f(X_i^t) \\ \\ X_i^t & \text{Otherwise} \end{cases} \tag{28}$$

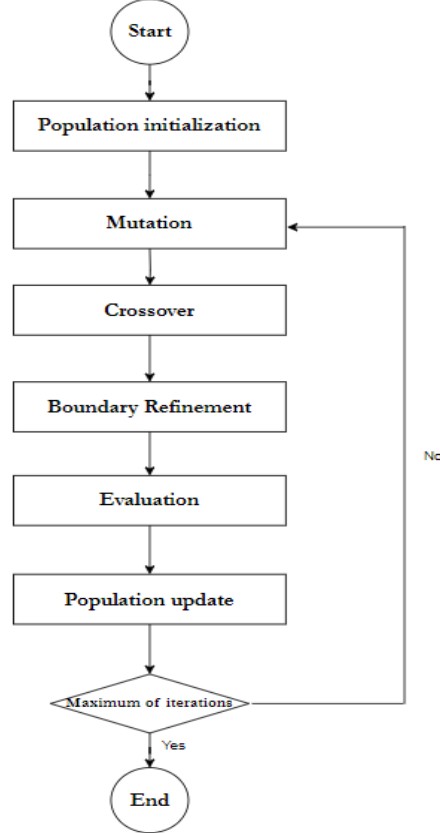

**Figure 2.** A representative scheme of the DE algorithm.

In this study, the population size was set to 100, CR = 0.8, SF = 0.2 and the maximum number of iterations was set to 100.

In our DCFC case, the process consisted of finding the values of the vector of the seven predefined unknown parameters $X = (\alpha_a, \alpha_c, \sigma_g, i_{0,c}, i_{l_{O_2}}, i_{l_{CO_2,c}}, i_{l_{CO_2,a}})$ that minimize the root mean square error (RMSE) between the model's predictions and the experimental data in order to fulfill the actual I-V and I-P characteristics:

$$\text{RMSE}(X) = \sqrt{\sum_{n=1}^{N} \frac{(V_{exp} - V_{mod})^2}{N}} \tag{29}$$

Indeed, the DE algorithm is an evolutionary optimization algorithm that iteratively explores the parameter space by generating new candidate solutions and evaluating their fitness based on RMSEs by minimizing the errors between the experimental and the estimated voltage values. The algorithm gradually refines the parameter values until a satisfactory convergence is achieved, yielding the best-fitted model to the experimental data. This optimization approach allows for robust and efficient parameter estimation, leading to a more accurate representation of the underlying relationship between the model and the experimental data.

## 3. Results and Discussion

DE's swiftness stems from its ability to efficiently search for global optima, achieved through the evolution of a candidate solution population via mutation, crossover and selection operations, making it a valuable resource for promptly addressing intricate optimization challenges. Hence, the DE algorithm was used here to predict the optimized unknown electrochemical kinetic parameters of a DCFC fueled with APC using a $CO_2$ model operating at 923 K: $X_1 = (\alpha_a, \alpha_c, \sigma_g, i_{0,a}, i_{0,c}, i_{l_{O_2}}, i_{l_{CO_2,c}}, i_{l_{CO_2,a}})$ and using a $CO/CO_2$ model at two different temperatures T = 923 K: $X_2 = (\alpha_a, \alpha_c, \sigma_g, i_{0,a}, i_{0,c}, i_{l_{O_2}}, i_{l_{CO_2,c}}, i_{l_{CO_2,a}})$ and T = 1023 K: $X_3 = (\alpha_a, \alpha_c, \sigma_g, i_{0,a}, i_{0,c}, i_{l_{O_2}}, i_{l_{CO_2,c}}, i_{l_{CO_2,a}}, i_{l_{CO}})$ for comparison. The DE algorithm was also applied to a DCFC fueled with ASB to determine the optimal values of the unknown electrochemical kinetic parameters: $X = (\alpha_a, \alpha_c, \sigma_g, i_{0,c}, i_{l_{O_2}}, i_{l_{CO_2,c}}, i_{l_{CO_2,a}})$ for three operating temperatures: 873 K, 923 K and 973 K.

As stated in the previous section, we configured the parameters of the DE algorithm as follows: Population size (Pop) = 100, Crossover Rate (CR) = 0.8, Scale Factor (SF) = 0.2 and a maximum of 100 iterations (max_iter). The simulations were performed within a Jupyter Notebook using Python 3 software in an Anaconda Navigator employing the Differential Evolution (DE) algorithm to swiftly tackle optimization tasks; they were programmed on a Core(TM) i7-10750H CPU @ 2.60 GHz processor with 16 GB of RAM.

### 3.1. Parameters Optimization of $CO_2$ and $CO/CO_2$ DCFC Models Fueled with APC

The design and the operating data of the DCFC system can be found in references [10,32]. The cell voltages calculated using DE are compared with the experimental data reported by Chen et al. [32] for the $CO_2$ model and with the analytical data of Elleuch et al. [10] for the $CO/CO_2$ model. Table 1 outlines the scope of the optimization searches for the unknown kinetic parameters in the $CO_2$ producing model at 923 K.

**Table 1.** Setting boundaries for $CO_2$ model at 923 K.

| Parameter | $\alpha_a$ | $\alpha_c$ | $i_{oc}$ (A·m$^{-2}$) | $\sigma_g$ (S·m$^{-1}$) | $i_{l_{O_2}}$ (A·m$^{-2}$) | $i_{l_{CO_2,c}}$ (A·m$^{-2}$) | $i_{l_{CO_2,a}}$ (A·m$^{-2}$) |
|---|---|---|---|---|---|---|---|
| Lower bound | 0 | 0 | $10^2$ | 10 | $4 \times 10^3$ | $4 \times 10^3$ | $4 \times 10^3$ |
| Upper bound | 1 | 1 | $10^3$ | $10^2$ | $10^4$ | $10^4$ | $10^4$ |

Elleuch et al. [10] achieved an average absolute deviation of approximately 4%, attributing the uncertainty to a charge transfer coefficient value of 0.5, which they set. To improve their results, they adopted this coefficient as a fitting parameter in their study.

Knowing that the charge transfer coefficients $\alpha_a$ and $\alpha_c$ differ between the anode and cathode, we treated them as variables, leading to a satisfactory RMSE value of $2.4157 \times 10^{-3}$, as presented in Table 2.

**Table 2.** Optimized parameters calculated using DE algorithm for $CO_2$ model using experimental data from [32].

| Parameter | $\alpha_a$ | $\alpha_c$ | $i_{oc}$ (A·m$^{-2}$) | $\sigma_g$ (S·m$^{-1}$) | $i_{IO_2}$ (A·m$^{-2}$) | $i_{ICO_2,c}$ (A·m$^{-2}$) | $i_{ICO_2,a}$ (A·m$^{-2}$) | RMSE |
|---|---|---|---|---|---|---|---|---|
| Value | 0.9157 | 0.2814 | 198.2247 | 18.9668 | 6087.3704 | $4 \times 10^3$ | 5064.0225 | $2.4157 \times 10^{-3}$ |

The $\alpha_a$ and $\alpha_c$ coefficients essentially quantify the degree to which electron transfer influences the electrochemical reactions at the anode and the cathode of a DCFC. The obtained values, shown in Table 2, reflect the asymmetry of the transition state of the anodic or cathodic electrochemical reactions in terms of electron transfer. Indeed, a value of 0.5 signifies a symmetrical reaction, while deviations from this value indicate asymmetry. According to Table 2, electron transfer seems to be more activated on the anode side than on the cathode side.

The optimal DE results from the $CO_2$ producing DCFC model were then compared with the experimental data [32]. Figure 3 shows the I-V and I-P characteristic curves, demonstrating a notable consistency between the experimental and calculated results.

The DE algorithm results were also validated with analytical data by using both the $CO_2$ and $CO/CO_2$ models at a fixed temperature of 923 K [10]. It can be observed in Table 3 that the experimental data for the $CO_2$ model differed from the analytical values given in Table 4.

The same boundaries were fixed for both cases, which explains the close RMSE values (of the order of $10^{-3}$) presented in Tables 2 and 5.

The values of $\alpha_a$ and $\alpha_c$ obtained from DE algorithm calculation, as shown in Table 5, also reflect the asymmetry of the transition state of the anodic and cathodic electrochemical reactions in terms of electron transfer.

The boundaries used for the $CO/CO_2$ model are detailed in Table 6, and the optimized parameter values are provided in Table 7. However, due to the increased complexity of the $CO/CO_2$ model compared to the simple $CO_2$ model, we noted a rise in the RMSE value to $1.7 \times 10^{-2}$.

As stated by Elleuch et al. [10], the mechanisms taking place on the anode side [36], driven by the presence of CO, play a significant role in enhancing the performance of the DCFC. This enhancement is evident in Figure 4, where an increase in power density is illustrated. Furthermore, at low current densities, the voltage obtained from the DE calculation are notably higher, as indicated in Table 8.

In order to show the effect of operating temperature on DCFC performance—a commonly studied factor—the $CO/CO_2$ model was executed for two temperatures: 923 K and 1023 K. The predefined boundaries for both temperatures were the same (Table 6).

Table 9 shows the optimized parameters for the $CO/CO_2$ model at 1023 K. The RMSE values for this temperature are close to that of 923 K (around the order of $10^{-2}$). Notably, Table 9 reflects an increase in the charge transfer coefficients, aligning well with findings from the existing literature [39–41]. In many electrochemical systems, the charge transfer coefficient tends to rise with temperature due to the increased thermal energy, promoting molecular motion and collisions between reacting species and the electrode surface, and thus enhancing the charge transfer process. The obtained values of transfer coefficients reflect the high asymmetry of the transition state of the anodic and cathodic electrochemical reactions in terms of electron transfer.

Moreover, the electrolyte conductivity decreased after raising the temperature from 923 K to 1023 K. This decrease can be attributed to the intensified thermal decomposition of carbonate ions, leading to the generation of more $CO_2$ and oxygen ions. Consequently, the availability of carbonate ions for ionic conduction diminished, resulting in reduced ionic

conductivity in the electrolyte. Additionally, Figure 5 reveals that the calculated voltage and power values at 1023 K were higher than those at 923 K, confirming the improved cell performance with temperature increase.

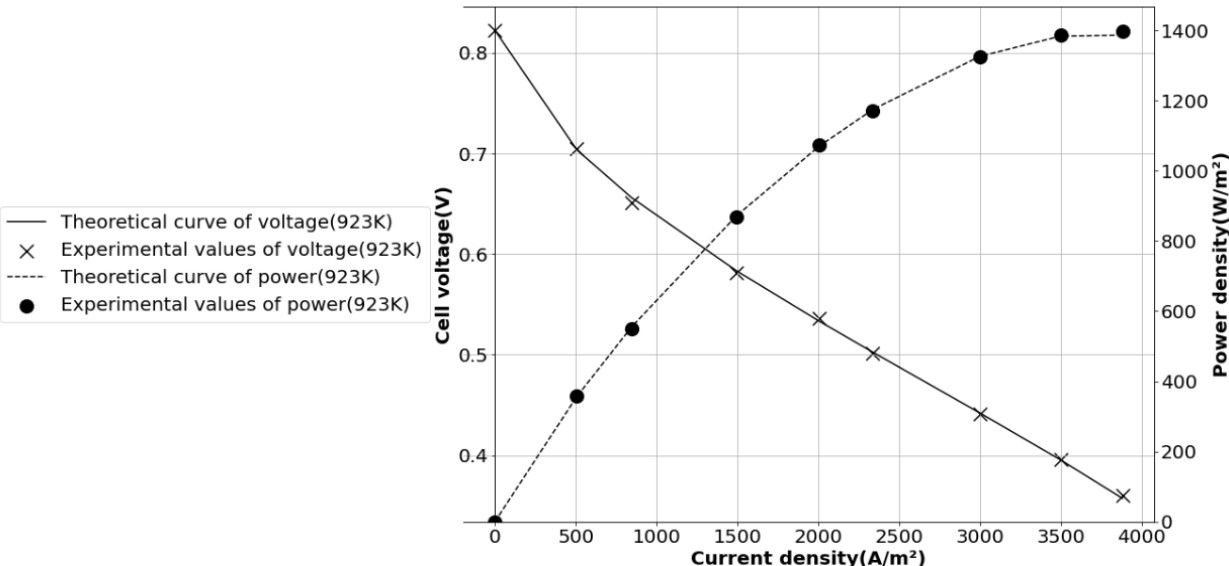

**Figure 3.** Validation of the DE algorithm results using the $CO_2$ model with experimental data at 923 K [32].

**Table 3.** DE voltage results for $CO_2$ model compared with experimental data of [32] at 923 K.

| i (A·m$^{-2}$) | Experimental Voltage (V) [32] | DE Results (V) |
|---|---|---|
| 0 | 0.8225 | 0.8225 |
| 507.1869 | 0.7047 | 0.7036 |
| 844.4271 | 0.6509 | 0.6563 |
| 1496.2882 | 0.5811 | 0.5832 |
| 2005.2647 | 0.5360 | 0.5334 |
| 2336.9177 | 0.5011 | 0.5025 |
| 3000.2965 | 0.4415 | 0.4420 |
| 3497.8281 | 0.3964 | 0.3955 |
| 3880.9128 | 0.3600 | 0.3572 |

**Table 4.** DE voltage results for $CO_2$ model compared with analytical data from [10] at 923 K.

| i (A·m$^{-2}$) | Analytical Voltage (V) [10] | DE Results (V) |
|---|---|---|
| 0 | 0.8223 | 0.8123 |
| 439.3939 | 0.6715 | 0.6829 |
| 878.7879 | 0.6254 | 0.6243 |
| 1318.1818 | 0.5885 | 0.5804 |
| 1757.5758 | 0.5515 | 0.5422 |
| 2196.9697 | 0.5146 | 0.5062 |
| 2628.7879 | 0.4731 | 0.4704 |
| 3507.5758 | 0.4292 | 0.4304 |
| 3068.1818 | 0.3692 | 0.3794 |
| 3946.9697 | 0.2862 | 0.2750 |

**Table 5.** Optimized parameters calculated using DE algorithm for $CO_2$ model using analytical data from [10].

| Parameter | $\alpha_a$ | $\alpha_c$ | $i_{oc}$ (A·m$^{-2}$) | $\sigma_g$ (S·m$^{-1}$) | $i_{lO_2}$ (A·m$^{-2}$) | $i_{lCO_2,c}$ (A·m$^{-2}$) | $i_{lCO_2,a}$ (A·m$^{-2}$) | RMSE |
|---|---|---|---|---|---|---|---|---|
| Value | 0.3392 | 0.2756 | 114.8181 | 36.6461 | $4 \times 10^3$ | 8581.8031 | 4074.0127 | $7.6834 \times 10^{-3}$ |

**Table 6.** Setting boundaries for CO/ $CO_2$ model at 923 K.

| Parameter | $\alpha_a$ | $\alpha_c$ | $i_{oa}$ (A·m$^{-2}$) | $i_{oc}$ (A·m$^{-2}$) | $\sigma_g$ (S·m$^{-1}$) | $i_{lCO_2,a}$ (A·m$^{-2}$) | $i_{lCO}$ (A·m$^{-2}$) | $i_{lO_2}$ (A·m$^{-2}$) | $i_{lCO_2,c}$ (A·m$^{-2}$) |
|---|---|---|---|---|---|---|---|---|---|
| Lower bound | 0 | 0 | 15 | 15 | 10 | $4 \times 10^3$ | $4 \times 10^3$ | $4 \times 10^3$ | $4 \times 10^3$ |
| Upper bound | 1 | 1 | 200 | 200 | $10^2$ | $10^4$ | $10^4$ | $10^4$ | $10^4$ |

**Table 7.** Optimized parameters calculated using DE algorithm for CO/ $CO_2$ model using analytical data from [10] at 923 K.

| Parameter | $\alpha_a$ | $\alpha_c$ | $i_{oa}$ (A·m$^{-2}$) | $i_{oc}$ (A·m$^{-2}$) | $\sigma_g$ (S·m$^{-1}$) | $i_{lCO_2,a}$ (A·m$^{-2}$) | $i_{lCO}$ (A·m$^{-2}$) | $i_{lO_2}$ (A·m$^{-2}$) | $i_{lCO_2,c}$ (A·m$^{-2}$) | RMSE |
|---|---|---|---|---|---|---|---|---|---|---|
| Value | 0.9762 | 0.6961 | 15 | 15 | 65 | 4368.6 | 4644.6 | 5849.9 | 7303.3 | $1.7 \times 10^{-2}$ |

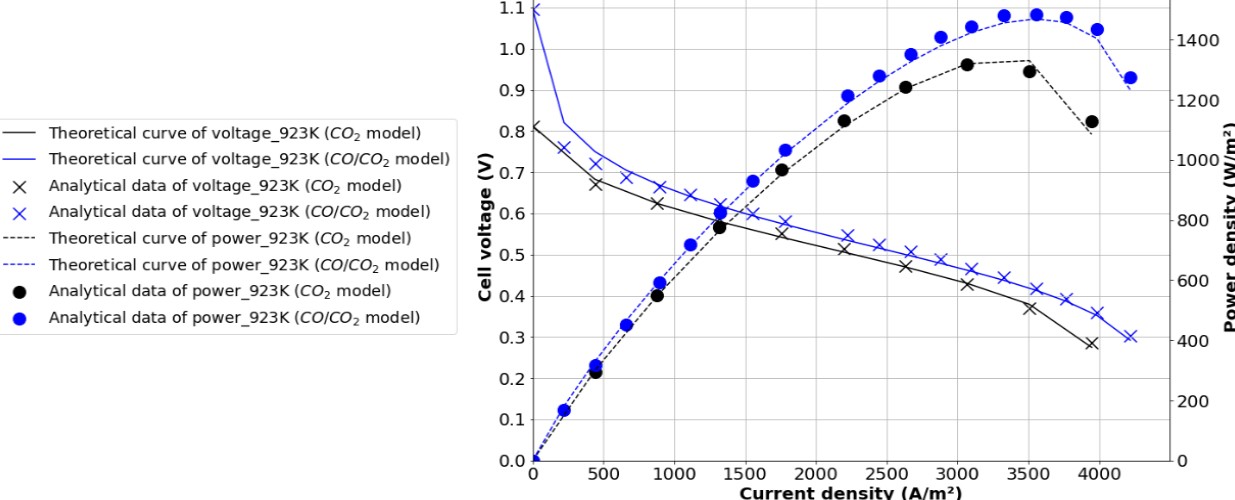

**Figure 4.** Validation of the DE algorithm results using the $CO_2$ and CO/$CO_2$ models with analytical data at 923 K [10].

**Table 8.** DE voltage results for CO/ $CO_2$ model compared with analytical data from [10] at 923 K.

| i (A·m$^{-2}$) | Analytical Voltage (V) [10] | DE Results (V) |
|---|---|---|
| 0 | 1.0962 | 1.0962 |
| 219.6970 | 0.7612 | 0.8210 |
| 439.3939 | 0.7223 | 0.7502 |
| 659.0909 | 0.6877 | 0.7053 |
| 893.9394 | 0.6646 | 0.6691 |
| 1113.6363 | 0.6461 | 0.6411 |
| 1325.7576 | 0.6231 | 0.6174 |
| 1553.0303 | 0.6000 | 0.5945 |
| 1780.3030 | 0.5815 | 0.5733 |
| 2219.6970 | 0.5470 | 0.5350 |

**Table 8.** *Cont.*

| i (A·m$^{-2}$) | Analytical Voltage (V) [10] | DE Results (V) |
|---|---|---|
| 2446.9700 | 0.5238 | 0.5159 |
| 2666.6667 | 0.5077 | 0.4974 |
| 2878.7879 | 0.4892 | 0.4792 |
| 3098.4849 | 0.4662 | 0.4595 |
| 3325.7576 | 0.4454 | 0.4378 |
| 3553.0303 | 0.4177 | 0.4136 |
| 3765.1515 | 0.3923 | 0.3872 |
| 3984.8485 | 0.3600 | 0.3524 |
| 4219.6970 | 0.3023 | 0.2922 |

**Table 9.** Optimized parameters calculated using DE algorithm for $CO/CO_2$ model using analytical data from [10] at 1023 K.

| Parameter | $\alpha_a$ | $\alpha_c$ | $i_{oa}$ (A·m$^{-2}$) | $i_{oc}$ (A·m$^{-2}$) | $\sigma_g$ (S·m$^{-1}$) | $i_{lCO_2,a}$ (A·m$^{-2}$) | $i_{lCO}$ (A·m$^{-2}$) | $i_{lO_2}$ (A·m$^{-2}$) | $i_{lCO_2,c}$ (A·m$^{-2}$) | RMSE |
|---|---|---|---|---|---|---|---|---|---|---|
| Value | 1 | 0.9747 | 15 | 27.965 | 41.485 | 4000 | 6396.861 | 4000 | 7116 | $1.087 \times 10^{-2}$ |

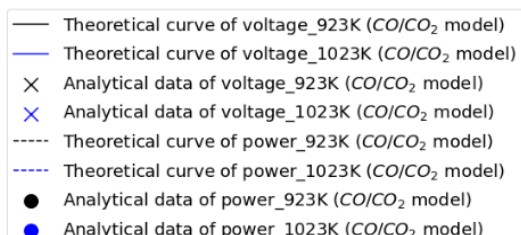
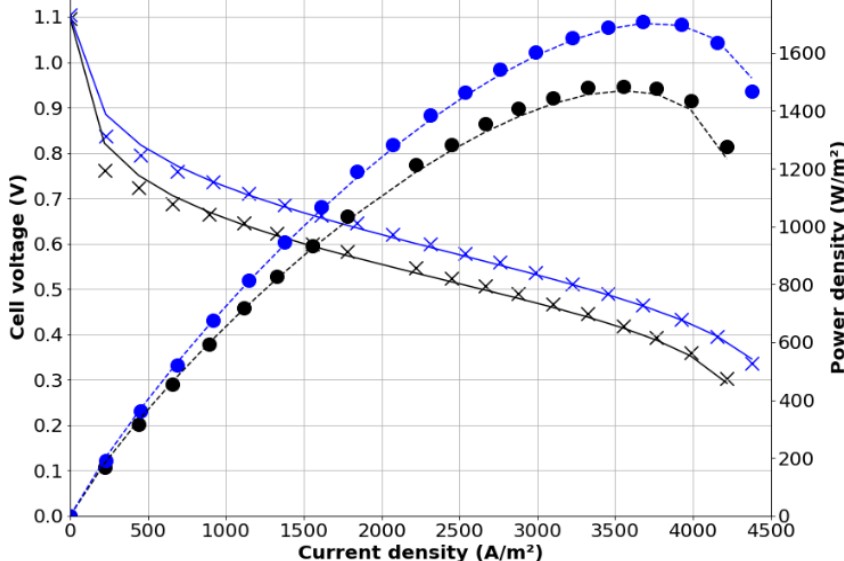

**Figure 5.** Validation of the DE algorithm results using the $CO/CO_2$ model with analytical data at 923 K and 1023 K [10].

### 3.2. Parameter Optimization of CO/CO$_2$ DCFC Models Fueled with ASB

After the successful application of the $CO/CO_2$ model on a APC-fueled DCFC, our focus shifted to applying it on a DCFC fueled with almond shell biochar (ASB). The design and the operating data of the DCFC system can be found in references [5,6]. The model validation for ASB was conducted with predefined boundaries, as shown in Table 10. Table 11 shows the high asymmetry of the transition state of electrode electrochemical reactions in terms of electron transfer at 823 K and 923 K with a relative symmetry at 973 K. The increase in ($i_{l_{CO,a}}$) and the decrease ($i_{l_{CO_2,a}}$) at the anode side, with almost stable ($i_{l_{CO_2,c}}$) and ($i_{l_{O_2}}$) at the cathode side when the DCFC temperature increases, shows that the $CO/CO_2$ model well describes the electrochemical mechanism in the case of DCFCs fueled with ASB.

**Table 10.** Setting boundaries for DCFCs fueled with almond shell using $CO/CO_2$ model (3500 is taken as lower bound for the characteristic currents at 923 K).

| Parameter | $\alpha_a$ | $\alpha_c$ | $i_{oa}$ (A·m$^{-2}$) | $i_{oc}$ (A·m$^{-2}$) | $\sigma_g$ (S·m$^{-1}$) | $i_{ICO_2,a}$ (A·m$^{-2}$) | $i_{ICO}$ (A·m$^{-2}$) | $i_{IO_2}$ (A·m$^{-2}$) | $i_{ICO_2,c}$ (A·m$^{-2}$) |
|---|---|---|---|---|---|---|---|---|---|
| Lower bound | 0 | 0 | 1 | 1 | 1 | $4 \times 10^3$ | $4 \times 10^3$ | $4 \times 10^3$ | $4 \times 10^3$ |
| Upper bound | 1 | 1 | $10^3$ | $10^3$ | $10^3$ | $10^4$ | $10^4$ | $10^4$ | $10^4$ |

**Table 11.** Optimized parameters calculated using DE algorithm for $CO/CO_2$ model using experimental data from [6].

| Parameter | $\alpha_a$ | $\alpha_c$ | $i_{oa}$ (A·m$^{-2}$) | $i_{oc}$ (A·m$^{-2}$) | $\sigma_g$ (S·m$^{-1}$) | $i_{ICO_2,a}$ (A·m$^{-2}$) | $i_{ICO}$ (A·m$^{-2}$) | $i_{IO_2}$ (A·m$^{-2}$) | $i_{ICO_2,c}$ (A·m$^{-2}$) | RMSE |
|---|---|---|---|---|---|---|---|---|---|---|
| Value (823 K) | 0.6255 | 0.0328 | 495.2078 | 1000 | 695.4676 | 9345.5575 | 7895.6720 | 6524.9075 | 5360.3827 | $3.14 \times 10^{-2}$ |
| Value (923 K) | 0.7536 | 0.9241 | 356.6782 | 476.6354 | 2.6200 | 7593.2842 | 7581.9700 | 4371.4425 | 8439.4424 | $4.77 \times 10^{-2}$ |
| Value (973 K) | 0.4073 | 0.5671 | 787.6188 | 1000 | 8.3190 | 4000 | 8961.5211 | 6972.4070 | 9341.9678 | $8.10 \times 10^{-2}$ |

It is noteworthy that the RMSE values increased compared to those obtained in the APC-fueled DCFC, but they remained of the order of $10^{-2}$. As shown in Figure 6, the fuel cell's performance was, once again, enhanced with an increase in temperature [39–41].

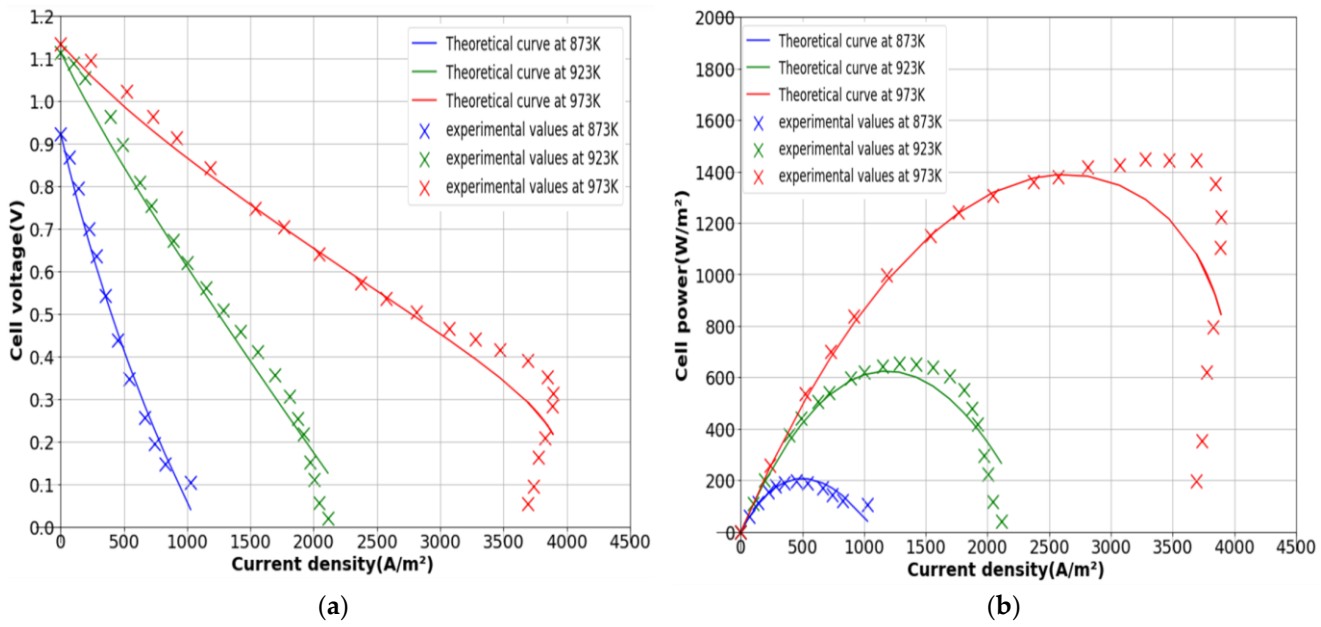

(**a**)          (**b**)

**Figure 6.** Validation of DE algorithm using $CO/CO_2$ model by comparison with experimental data [6] at different operating temperatures: (**a**) I-V Characteristic; (**b**) I-P Characteristic.

## 4. Conclusions

The Differential Evolution (DE) algorithm allowed for the prediction and the optimization of key kinetic parameters of DCFCs using $CO_2$ and $CO/CO_2$ electrochemical models. The DE was firstly used to analyze a DCFC powered by APC and subsequently applied to an ASB-fueled DCFC, showing the asymmetry of the transition state of the anodic and cathodic electrochemical reactions in terms of electron transfer in both cases.

The computational outcomes closely aligned with the experimental data, particularly for the APC-powered DCFC in both model scenarios. The validation results for the ASB-fueled DCFC were also promising, with a slight increase in the Root Mean Square Error

(RMSE). This discrepancy can be attributed to the ASB's complex structure and chemical composition, which is not accounted for in the $CO/CO_2$ model, as well as the concentration polarization limitations. This study delved into the intricate electrochemical mechanisms within DCFCs fueled with non-conventional fuel; the DE algorithm was successfully used and achieved satisfying accuracy in solving such a nonlinear problem, showing good agreement with the analytical models and experimental data.

**Author Contributions:** Conceptualization, K.H.; methodology, A.C. and K.H.; software, A.C.; validation, A.C. and K.H.; formal analysis, A.C. and K.H.; investigation, A.C. and K.H.; resources, K.H.; data curation, A.C. and K.H.; writing—original draft preparation, A.C.; writing—review and editing, K.H.; supervision, K.H. All authors have read and agreed to the published version of the manuscript.

**Funding:** This research received no external funding.

**Institutional Review Board Statement:** Not applicable.

**Informed Consent Statement:** Not applicable.

**Data Availability Statement:** Data sharing within this article.

**Conflicts of Interest:** The authors declare no conflicts of interest.

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
