# Peer review of "Artificial Intelligence for Electrochemical Prediction and Optimization of Direct Carbon Fuel Cells Fueled with Biochar"

_2673-3293, doi:10.3390/electrochem5010002_

Round 1
Reviewer 1 Report
Comments and Suggestions for Authors
Dear authors,
thank you for the manuscript. From my point of view it requires major revision, especially due to the following reasons:
- Please optimise the figure in a way that the font size correpsonds to the font size of the text.
- It is not required to present redundant results in figures and table.
- It would make sense to describe the operating principle of a DCFC in more detail in order to develop the discussion based on that description. It would also be helpful in the context of comparing modelling results with experimental results.
Comments on the Quality of English LanguageFrom a language point of view please revise some of the expressions in order to achieve more clarity.
Author Response
Authors thank very much the Reviewer for his valuable comments to improve the quality of the revised manuscript. Please find attached our point by point reply to your comments.

Reviewer 2 Report
Comments and Suggestions for Authors
1. The manuscript should be carefully checked before submission. There are some errors in the figures, such as 2 (b) in Figure 3.
2. Figures can be optimised to be easy to read.
3. Some tables can be included in supplementary material.
4. The highlights of this paper are not emphasised.
5. The flow of the manuscript should be improved to be more logical.
Comments on the Quality of English LanguageThe overall quality of English is good, but some expressions could be polished.
Author Response

(The authors gave the same response as above.)

Reviewer 3 Report
Comments and Suggestions for Authors
The introduction leans heavily on the author's own work with limited reference to external sources. To provide a more balanced and comprehensive view, it would be beneficial to include a broader range of external citations in the literature review DCFC.
The introduction should strive to clarify several key aspects: First, it should explicitly outline what is missing in the current state of research within this field, shedding light on the specific research gaps. Secondly, it should provide insight into the existing optimization efforts in the area, detailing what prior work has been done and highlighting any notable achievements. Lastly, it should articulate the needs and expectations of the research community with regard to the utilization of the Differential Evolution (DE) algorithm for optimization.
While the paper successfully presents optimized parameters, it falls short in connecting these values to their real-world applicability. Readers would benefit from a more explicit linkage between the optimized parameters and their significance in practical applications.
The quality of the figures used in the paper requires improvement, as they presently lack readability and clarity. Enhancing the visual representation of the data will enhance the overall presentation and understanding of the research.
There is a missing of a literature reference in line 371, which should be addressed for proper citation and source attribution.
To strengthen the paper's overall contribution, it is advisable to provide a more critical and nuanced assessment of how the DE algorithm benefits the research area. While it is evident that the DE algorithm aligns with experimental data and references, a more comprehensive analysis should explore how this methodology can make a substantial and distinctive contribution to the field of study.
Author Response

(The authors gave the same response as above.)

Round 2
Reviewer 1 Report
Comments and Suggestions for Authors
Thank you for the reviewed manuscript and considering all the remarks made.
Author Response
Authors thank very much the Reviewer for his efforts and valuable recommendations to improve the quality of the paper.

Reviewer 3 Report
Comments and Suggestions for Authors
Thank you for addressing the comments. Your responses have clarified key points, and I appreciate the improvements made.
Author Response
Authors thank very much the Reviewer for his valuable comments and recommendations to improve the quality of the paper.
